# Zearalenone Induces Apoptosis and Autophagy in a Spermatogonia Cell Line

**DOI:** 10.3390/toxins14020148

**Published:** 2022-02-17

**Authors:** Ran Lee, Dong-Wook Kim, Won-Young Lee, Hyun-Jung Park

**Affiliations:** 1Department of Stem Cell and Regenerative Biology, Konkuk University, 1 Hwayang-dong, Gwangjin-gu, Seoul 05029, Korea; ranran2424@gmail.com; 2Department of Swine & Poultry Science, Korea National College of Agriculture and Fisheries, 1515, Kongjwipatjwi-ro, Deokjin-gu, Jeonju-si 54874, Jeollabuk-do, Korea; poultry98@korea.kr; 3Department of Beef & Dairy Science, Korea National College of Agricultures and Fisheries, 1515, Kongjwipatjwi-ro, Deokjin-gu, Jeonju-si 54874, Jeollabuk-do, Korea; leewy81@korea.kr; 4Department of Animal Biotechnology, Sangji University, 83, Sangjidae-gil, Wonju-si 26339, Gangwon-do, Korea

**Keywords:** zearalenone, GC-1 spg cells, reproductive toxicity, apoptosis, autophagy

## Abstract

Zearalenone (ZEN), a widely known mycotoxin, is mainly produced by various *Fusarium* species, and it is a potent estrogenic metabolite that affects reproductive health in livestock and humans. In this study, the molecular mechanisms of toxicity and cell damage induced by ZEN in GC-1 spermatogonia (spg) cells were evaluated. Our results showed that cell viability decreased and apoptosis increased in a dose-dependent manner when GC-1 spg cells were exposed to ZEN. In addition, the key proteins involved in apoptosis, cleaved caspase-3 and -8, BAD, BAX, and phosphorylation of p53 and ERK1/2, were significantly increased in ZEN-exposed GC-1 spg cells for 24 h, and cytochrome c was released from mitochondria by ZEN. Interestingly, ZEN also triggered autophagy in GC-1 spg cells. The expression levels of the autophagy-related genes Atg5, Atg3, Beclin 1, LC3, Ulk1, Bnip 3, and p62 were significantly higher in ZEN-treated GC-1 spg cells, and the protein levels of both LC3A/B and Atg12 were remarkably increased in a dose-dependent manner in ZEN-exposed GC-1 spg cells compared to the control. In addition, immunostaining results showed that ZEN-treated groups showed a remarkable increase in LC 3A/B positive puncta as compared to the control in a dose-dependent manner based on confocal microscopy analysis in GC-1 spg cells. Our findings suggest that ZEN has toxic effects on tGC-1 spg cells and induces both apoptosis and autophagy.

## 1. Introduction

Zearalenone (ZEN), also known as the F-2 toxin, is a non-steroid mycotoxin with estrogenic effects produced by some *Fusarium* and *Gibberella* species [1]. ZEN is mainly found in contaminated grains such as oats, corn, wheat, sorghum, and rice, but also in food and feed that are improperly stored [2,3]. Acute and chronic toxicity to both humans and animals is a major concern worldwide. In addition, ZEN can accumulate in the body and induce reproductive toxicity. ZEN derivatives act similarly to 17β-estradiol (E_2_) by inhibiting steroid hormones because ZEN can bind to the estrogen receptor, and these complexes trigger estrogen response elements (EREs) [4]. In vivo studies have reported on the toxicological effects of ZEN in females, including functional alterations in reproductive organs, a depressing of the maturation of ovarian follicles, oocyte death in the follicles, a lack of ovulation [5,6], decreased fertility, increased embryo-lethal resorption, and abnormal hormone levels in the female reproductive system of rodents and livestock [7,8]. In male rodents, testes consist of seminiferous tubules and interstitial areas, which mainly contain Sertoli cells, Leydig cells, and various germinal cells [9]. Male mice exposed to this mycoestrogen showed an increased number of abnormal spermatozoa; additionally, a low pregnancy rate was observed when females were mated with ZEN-exposed males [10]. ZEN can also reduce acrosome reactions [11], the ability to bind spermatozoa and zona pellucida [12], sperm average path velocity (VAP), and straight-line velocity (VSL). Pang et al. reported that DNA double-strand breaks were prevalent in spermatogenic cells after 28 days of ZEN stimulation, which resulted in apoptosis of sperm cells [13]. In addition, Yang et al. reported that ZEN suppressed testosterone secretion and the expression levels of transcription of 3beta-HSD-1, P450scc, and StAR, which are related to steroidogenesis in testicular Leydig cells [14]. Similarly, a disruption of Leydig cell development and steroidogenesis was also observed in ZEN-exposed rats in an in vivo study [15]. Several studies have also investigated the effects of ZEN on testicular Sertoli cells. ZEN altered the cytoskeletal structure via the oxidative stress-autophagy-ER stress pathway in mouse TM4 Sertoli cells [16] and induced apoptotic cell death in TM4 cells [17]. ZEN also disrupts the development of animal embryos mainly in the role of the placenta on ZEN transmission, pregnancy rate and follicular integrity of offspring, and mammary gland development in neonatal mice [18]. In addition, ZEN modulates the activity of both testes and ovaries in mice, swine, and cows [19]. ZEN intake affected ovarian antral follicles, increased infertility, and hyperestrogenism in cows [20,21]. ZEN-exposed gilts showed an increased organ size, an inducement of hyperplasia in the smooth muscles of the corpus uteri, and a decrease in the number of follicles in the cortex and apoptotic cells in the ovaries [22].

The molecular mechanisms underlying the toxic effects of the major mycotoxins have been established, and several mycotoxins induced oxidative stress in cells via ROS generation and stimulate inflammatory reactions [23,24]. ZEN also damages the antioxidant defense system and suppresses glutathione peroxidase (GPx), superoxide dismutase (SOD), and catalase (CAT) in mouse testes [25]. In addition, ZEN has previously upregulated intracellular ROS levels, which disrupted the balance of ER homeostasis and ER stress, and finally induced cell apoptosis through the ATP/AMPK pathway in testicular Sertoli cells [25,26]. Chen et al. observed ZEN-induced apoptosis of ovarian granulosa cells through autophagy activation and apoptotic pathway inhibition [27].

However, the molecular mechanism of ZEN toxicity in mouse-derived GC-1 spg cells, known as spermatogonia cells, has not been studied. Therefore, we investigated the detailed molecular mechanism underlying ZEN-mediated toxicity in GC-1 spg cells.

## 2. Results

### 2.1. Zearalenone Reduces Cell Viability in the GC-1 spg Spermatogonia Cell Line

Several studies have reported that ZEN induces male germ cell apoptosis in rodents through an in vivo system [28,29]. Based on these results, we evaluated the effects of ZEN in vitro using spermatogonia GC-1 spg cells, which were derived from 10-day mouse testes. Cell viability assays were performed in GC-1 spg cells following exposure to ZEN to determine the cytotoxic effects of ZEN. The cell viability of GC-1 spg cells decreased in a dose-dependent manner when the cells were exposed to 30–300 µM ZEN for 24 h compared to the control (0 µM ZEN). According to these results, the LC_50_ values were 54.53 μM (Figure 1A), and concentrations of 10–100 μM ZEN were used for this study.

Following ZEN treatment for 24 h, cellular morphology was observed under light microscopy, and ZEN clearly inhibited cell proliferation (Figure 1B). Based on these results, we examined apoptosis via TUNEL staining. Figure 1C shows that the nuclei of apoptotic cells exhibited red fluorescence against a dark background, and apoptotic cells were clearly increased by ZEN treatment in GC-1 spg cells compared to the control.

Specifically, the percentage of TUNEL-positive cells was significantly higher in the 50–100 nM ZEN treated groups than in the controls (Figure 1D).

### 2.2. Zearalenone Induces the Expression of Pro-Apoptotic Proteins in GC-1 Spg Cells

Next, we dissected the mechanism of ZEN-induced apoptosis in GC-1spg cells. The expression levels of key proteins involved in apoptosis such as cleaved caspase-3 and -8, BAD, BAX, phospho p53 and ERK1/2 were observed in GC-1 spg cells treated with ZEN at different doses (0–100 μM) (Figure 2). The levels of BAX and BAD proteins were increased, and activated caspase-3 and caspase-8 were dose-dependently enhanced in ZEN-exposed GC-1 spg (Figure 2A–D). MAPK proteins have been reported to be common apoptotic signaling molecules [30]. Our results also showed that treatment of GC-1 spg with ZEN resulted in increased p53 and ERK1/2 phosphorylation (Figure 2E,F). Specifically, the expression of all these proteins was significantly higher in 50–100 μM ZEN exposed GC-1 spg than in the control.

### 2.3. Zearalenone Triggers Apoptosis via Mitochondrial Pathway

Cytochrome c is released from the mitochondria to the cytosol, where it participates in caspase activation [31]. Therefore, we investigated whether ZEN could induce the release of cytochrome c in GC-1 spg cells. The expression and localization of cytochrome c protein in GC-1 spg cells were observed using immunoblotting and confocal immunofluorescence microscopy, respectively. In Figure 3A, a strong expression of cytochrome c, which is red fluorescence in 50–10 μM ZEN-treated GC-1 spg cells, can be observed; this expression demonstrates a more diffuse localization pattern in ZEN-treated cells compared to the controls. Cytochrome c was redistributed in the region surrounding the lamin A/C+ nucleus envelope in ZEN-treated cells (Figure 3A). The protein levels of cytochrome c in ZEN-treated cells also consistently increased in a dose-dependent manner (Figure 3B).

### 2.4. Zearalenone Triggers Autophagy in GC-1 spg Cells

A recent study reported ZEN-induced autophagy in TM4 Sertoli testicular cells [32]. Based on these studies, we examined whether ZEN induced autophagy in GC-1 spg cells, even though ZEN induced apoptotic cell death. Our results showed that the expressions of autophagy-related genes such as Atg5, Atg3, Beclin1, LC3, Ulk1, Bnip3, and p62 significantly increased in 100 μM ZEN-treated GC-1spg cells compared to the control (Figure 4A). Additionally, we confirmed the expression levels of autophagy-related proteins. Immunostaining of LC3A/B revealed that ZEN-treated groups showed a remarkable increase in LC 3A/B positive puncta compared to the control in a dose-dependent manner by confocal microscopy analysis in GC-1spg cells (Figure 4B). The expression levels of both Atg12 and LC3A/B proteins were consistently higher in ZEN-treated groups than in the control in a dose-dependent manner, according to immunoblotting analysis (Figure 4C). These data suggest that ZEN can trigger autophagy in GC-1 spg cells as well as apoptotic cell death.

## 3. Discussion

Mycotoxins such as aflatoxins, zearalenone, ochratoxin, and deoxynivalenol negatively affect the reproductive system in vertebrates, including livestock and humans.

Although the effects of ZEN on male germ cells in the testes were studied through in vivo studies, the detailed mechanism of its toxicity has not been fully investigated in in vitro cultured spermatogonia [28]. Spermatogenesis is a continuous process in which early germ cells undergo mitosis, meiosis [33], and the formation of mature sperm. Spermatogenesis disorder induces male infertility, which is reflected by a decreased sperm number and motility, and an increased deformity rate [34]. GC-1 spg cells are type B spermatogonia, which undergo growth and become primary spermatocytes that undergo meiosis [34].

In this study, we have demonstrated that ZEN induced both apoptosis and autophagy in in vitro cultured GC-1 spg cells. TUNEL-labeled GC-1 spg cells induced by 50–100 nM ZEN significantly increased with an increase in treatment dose. Similar to our study, Kim et al. reported that degenerating stages I-VI germ cells underwent apoptosis by TUNEL when 10-week-old rats were treated with an intraperitoneal (i.p.) dose of 5 mg/kg ZEN. Spontaneous apoptotic cells were found in both spermatogonia and spermatocytes of stage I-VI and XII-XIV seminiferous tubules [28]. However, Kim et al. did not describe the detailed mechanism of apoptotic cell death in spermatogonia. Accordingly, our results show a major signaling pathway for apoptosis in GC-1 spg cells. The expression levels of key apoptotic proteins such as cleaved caspase-3 and -8, BAD, and BAX obviously increased in ZEN-exposed GC-1 spg cells. Similarly, several studies reported increases in the apoptosis-promoting gene BAX and Caspase-3 activity, and that the expression of the inhibitory gene BCL2 decreased in ZEN-treated bovine MAC-T cells [35], chicken granulosa cells [36], and human leukemic cells such as U937 and HL-60 [37]. In addition, our results showed an increase in the levels of phosphorylated p53 and ERK 1/2 in GC-1 spg cells following ZEN treatment. Lee et al. also reported ZEN-induced apoptosis of endothelial cells through an ERK1/2/p53/caspase-3 signaling pathway, which was similar to our results [38]. ERK activation mediates apoptosis following DNA damage, and p53 is phosphorylated on multiple residues in the amino- and carboxy-terminal domains by several different protein kinases when exposed to DNA damage stress. In addition, ERK-mediated phosphorylation of p53 has been thoroughly studied in several experimental systems, including in resveratrol-exposed epidermal cells [39] and cisplatin-exposed ovarian cells [40]. The results of our study also showed that ZEN increased cytochrome c protein expression. Cytochrome c is a key molecule in mitochondria-mediated apoptosis, and the permeabilization of the mitochondrial outer membrane during apoptosis is critical for the intrinsic pathway of cell death. Furthermore, a caspase-dependent apoptotic mechanism is often initiated in response to the release of cytochrome c from the mitochondria into the cytosol [41]. Several studies have also shown that ZEN promotes the release of cytochrome c from the mitochondria to the cytoplasm in rat Sertoli cells [42], human leukemic cells [37], endometrial stromal cells [43], and hepatocytes [44]. There are major signaling pathways that can lead to cell death, including autophagy and apoptosis. Autophagy plays an important physiological role in cells and promotes both cell survival and cell death [45]. Although ZEN-induced apoptosis has been widely documented, the role of autophagy in ZEN-induced toxic responses has not been thoroughly understood in spermatogonia.

In our study, ZEN also induced autophagy in cultured GC-1 spg cells. Autophagy-related genes such as Atg5, Atg3, Beclin 1, Atg12, LC3, Ulk1, Bnip3, and p62, and the protein levels of Atg12 and LC31/II significantly increased in ZEN-treated GC-1 spg cells. Several studies have shown that mycotoxins induce autophagy by increasing the levels of LC3 II/GAPDH and Beclin 1. For example, aflatoxin induced autophagy in THP-1 monocyte and RAW264.7 macrophage cells [46], and fumonisin induced autophagic cell death in HepG2 and MARC-145 cells [47,48]. Ochratoxin A (OTA) significantly improved the expressions of LC3 II and Atg5 proteins in the kidneys and spleens of pigs [49]. Additionally, OTA can trigger mitochondrial dysfunction as well as apoptotic and autophagic cell death in human gastric epithelial cells [50]. The protein levels of LC3 II and Beclin 1 increased in cardiac cells, thereby demonstrating that autophagy acts as a protective response against ZEN toxicity [51]

Interestingly, our results showed that both apoptotic cell death and autophagy were observed in ZEN-exposed GC-1spg cells. Similarly, ZEN induced apoptosis by activating caspases and triggering autophagy in primary rat Leydig cells [52]. The connection between apoptosis and autophagy in the form of cell death has been described in another study [53]. For example, there are two ways in which either the process of apoptosis may control autophagy or the process of autophagy may control apoptosis.

Autophagy may be able to kill a cell by actively degrading cellular components, such as mitochondria, to the point that the cell can no longer survive. Nonetheless, this does not mean that the process of autophagy directly acts on the cell death machinery. Cells eventually lead to apoptosis through autophagy, which may degrade cellular components [54,55]. In contrast, several studies have demonstrated that autophagy prevents cell death and may initiate a survival mechanism to cellular stress, such as nutrient and growth factor deprivation [56]. As described previously, it is possible that there is an indirect connection between autophagy and apoptosis in ZEN-exposed GC-1 spg cells. Specifically, p62 is a critical player in the selective autophagic degradation of proteins such as mitochondria, and p62 interacts directly with several apoptotic and survival pathway proteins, including ERK and caspase-8 [57,58]. Similarly, our results also showed that the levels of cleaved caspase-8, phospho-ERK1/2 proteins and p62 were higher in ZEN-treated GC-1 spg cells than in the control. The interaction between caspase-8 and p62 is especially intriguing because p62 is important in ensuring the efficient activation of caspase-8 in response to death receptor activation [58,59]. Moreover, according to a recent study, caspase-8 is degraded via autophagy [60]. There is considerable evidence that crosstalk exists between autophagy-related proteins and apoptotic factors. Ozlem et al. reported that caspase-8 is directly responsible for Atg3 cleavage and suggested that Atg 3 provides a novel link between apoptosis and autophagy during receptor-activated cell death [61]. Another study reported that Atg12 conjugation to Atg3 regulates mitochondrial homeostasis and cell death [62]. All these previous studies support our results.

Obermski et al. reported that the levels of ZEN in the blood of gilts with clinical symptoms of toxicosis after being fed a diet that included low ZEN content. The concentration of 24.13 ± 5.98 ng/mL^−1^ α-ZEA was observed in immature hybrid gilts in the second hour of ZEN administration (400 μg/kg^−1^ body weight) [63]. The blood concentration of 24 ng/mL^−1^ α-ZEA can be expressed as approximately 1 μM, which is lower than the concentration of ZEN used in our in vitro experiment.

Additionally, Alternaria (AOH) and alternariol methyl ether (AME) are other estrogenic mycotoxins which were formed by *Alternaria* species. AOH forms reactive oxygen species (ROS) and trigger various DNA damage response pathway in mammalian cells [64,65]. Another major mycotoxin, i.e., the T-2 toxin, also induced both autophagy and apoptosis in liver cells. However, their results demonstrated that autophagy helps protect cells from T-2 toxin-induced apoptosis, which is contrary to our results. The expression of apoptosis-associated proteins, such as PARP-1 and caspase-3, peaked at 6 h, whereas the detection of autophagy-associated proteins such as p62 and Beclin 1 decreased at 6 h following exposure to the T-2 toxin [66]. In contrast, our results showed that both apoptosis and autophagy are highly conserved processes in ZEN-exposed GC-1 spg cells. Although the detailed mechanism of crosstalk between autophagy and apoptosis in ZEN-exposed GC-1 spg cells is unclear, our results have clearly demonstrated the toxic effect of ZEN on spermatogonia via the study of apoptosis and autophagy mechanisms. The results of this study are meaningful in that they have identified the toxic mechanism of ZEN in the early male germ cell line (GC-1 spg), which is most important for the male reproductive system.

## 4. Conclusions

In conclusion, our study showed that ZEN inhibits GC-1 spg cell proliferation by inducing cell death via both apoptosis and autophagy. This work suggested that ZEN can cause serious defects in male reproductive system through early germ cell death.

## 5. Materials and Methods

### 5.1. Cell Culture and Treatments

GC-1 spg was purchased from the Korean Cell Line Bank (KCLB 21715, Seoul, South Korea). The cells were then cultured in Dulbecco’s modified Eagle’s medium supplemented with 10% fetal bovine serum and 1% penicillin–streptomycin in a humidified atmosphere of 5% CO_2_ at 37 °C. ZEN (Sigma-Aldrich, St. Louis, MO, USA) was dissolved in dimethyl sulfoxide (DMSO) to prepare a 1 M stock solution and was diluted to the desired concentration using the cell culture medium prior to cell culturing.

### 5.2. Cell Viability Assay

The proliferation rate of GC1-spg cells was determined by conducting a 3-(4,5-dimethylthiazole-2-yl)-2,5-diphenyl tetrazolium bromide (MTT) assay using the EZ-Cytox Viability Assay Kit (Daeil Lab Services Co., Seoul, Korea, #EZ1000), following the manufacturer’s instructions; then we measured the half-maximal inhibitory concentration (IC_50_) of ZEN in GC-1 spg cells. For the cell viability assay, cells were seeded in 96-well plates at a density of 3 × 10^3^ cells per well in a complete growth medium. Twenty-four hours after seeding, the culture medium was replaced with fresh medium containing ZEN (0–100 μM) and cultured for another 24 h. A cell viability assay reagent was added and cultured for 60 min, and the plates were read at 490 nm using an Epoch spectrophotometer (Bio Tek, Winooski, VT, USA).

### 5.3. Apoptosis Measured by TUNEL Assay

A terminal deoxynucleotide transferase-mediated deoxy-UTP nick end labeling (TUNEL) assay was used to detect apoptosis in GC-1 spg cultures. Cells were cultured on glass slides for 16 h and exposed to ZEN (0, 10, 50, and 100 µM) for 24 h, and then fixed with 4% paraformaldehyde in PBS for 15 min at 24 °C. After washing, the cells were incubated in a permeabilization solution (0.1%Triton X-100 in PBS) for 2 min on ice. Samples were incubated in 50 μL of the TUNEL reaction mixture (Roche, Mannheim, Germany) at 37 °C for 60 min in a humidified chamber, and nuclei were stained with 1 µg/mL 6-diamidino-2-phenylindole (DAPI) in PBS. After rinsing the samples with PBS three times, they were analyzed using a fluorescence microscope (Olympus IX73, Tokyo, Japan).

### 5.4. Immunofluorescence Staining

GC-1 spg cells treated with 1–100 µM ZEN were fixed and permeabilized in 4% paraformaldehyde at 4 °C for 30 min. The samples were then incubated overnight at 4 °C in a humidified chamber with primary antibodies such as LC3A/B, lamin a/c, and cytochrome C (Cell Signaling Technology, Danvers, MA, USA) at a dilution of 1:200. Slides were washed four times in PBS and incubated with a solution containing secondary antibodies for 1 h (Texas Red or FITC-conjugated secondary antibody) at room temperature. Finally, the coverslips were washed twice with PBS, all samples were mounted in a mounting medium (Sigma-Aldrich), and images were obtained using confocal microscopy (Carl Zeiss, Oberkochen, Germany; LSM 800).

### 5.5. Isolation of RNA and Quantitative Real-Time PCR (qRT-PCR) Analysis

Total RNA extraction was performed using the RNeasy Mini Kit (Qiagen, Hilden, Germany) with on-column DNase treatment (Qiagen) according to the manufacturer’s protocols. Complementary DNA (cDNA) was synthesized using MMLV reverse transcriptase (MGmed, Seoul, Korea) with an oligo(dT)30 primer, according to the manufacturer’s instructions. cDNA was mixed with SYBR Green master mix (Bioneer, Daejeon, Korea), and 1 pM of each primer was used for qPCR. Primers were designed using Primer3 (http://frodo.wi.mit.edu (accessed on 23 March 2021)). qRT-PCR was performed as previously described [67] using a QuantStudio 1 real-time PCR system (Applied Biosystems). The cycle threshold values were normalized against *GAPDH* gene expression. Amplification was carried out via initial denaturation at 94 °C for 1 min, followed by 40 cycles of denaturation at 95 °C for 10 s, annealing at 57 °C for 10 s, as well as at 72 °C for 20 s. The primers used to detect the mouse transcripts are listed in Table 1.

### 5.6. Western Blotting

Western blotting was performed as previously described [68]. Total protein was collected using RIPA lysis buffer (Thermo Scientific™, Rockford, IL, USA) and a protease inhibitor cocktail (Roche, Rotkreuz, Switzerland). Protein samples (20 μg) were separated with 4–12% gradient SDS-PAGE gels, transferred to PVDF membranes using a transfer blotting system (Bio-Rad, Hercules, CA, USA), and then the PVDF membranes were incubated with primary antibodies in blocking buffer solution (TBS with 0.1% tween-20 (TBST) + 1% bovine serum albumin) overnight at 4 °C. The membrane was then washed with TBST and incubated for 1 h with secondary antibodies (anti-mouse/rabbit antibody) for 1 h at room temperature. The antibodies used in this study are shown in Table 2. Blots were visualized using Pierce ECL solution (Thermo Fisher Scientific, Rockford, IL, USA) and an X-ray film. Image J software was used to quantify protein expression. Protein levels were normalized to those of the internal control β-actin or the inactive form.

### 5.7. Statistical Analysis

All experiments were independently repeated at least three times, and the results were expressed as mean ± standard error values. Data were evaluated using one-way analysis of variance (ANOVA). All statistical analyses were conducted using the SPSS statistical package, version 15.0, for Windows (IBM Corp., Somers, NY, USA). * *p* < 0.05 and ** *p* < 0.01 were considered statistically significant.

## Figures and Tables

**Figure 1 toxins-14-00148-f001:**
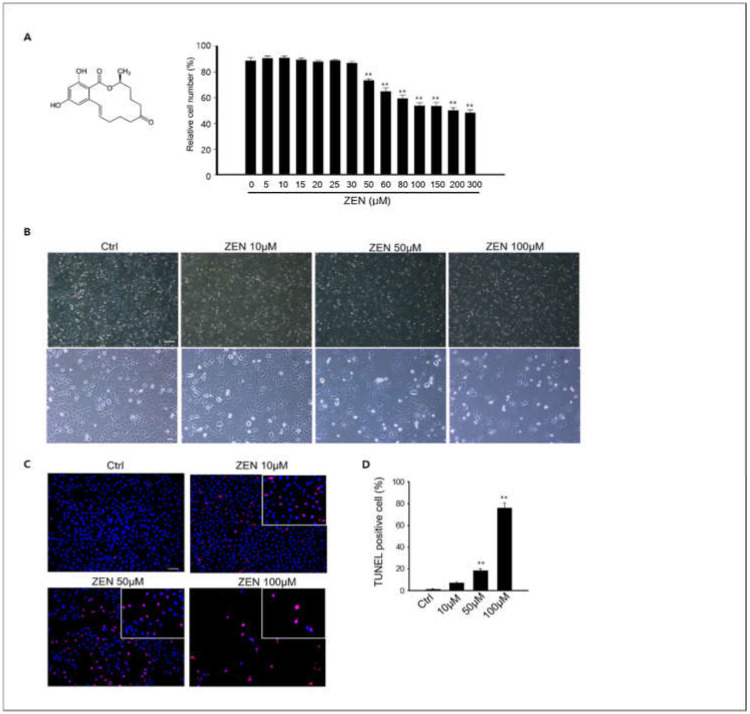
**Effects of ZEN on the viability of GC-1spg cells and apoptosis.** (**A**) A GC-1 spg cell viability assessment was performed using an MTT assay. The LC_50_ value of ZEN was 54.53 μM in GC-1 spg following a 24 h treatment with 0–300 μM. The differences between ZEN treatment groups were statistically significant compared to the 0 μM group. All data are presented as mean ± SD from three independent experiments (*n* = 3, ** *p* < 0.01). (**B**) The morphology of GC-1 spg cells following exposure to ZEN for different doses (0, 10, 50, and 100 μM) was examined and photographed using an inverted light microscope. Scale bars = 100 μm. (**C**) Determination of ZEN-induced apoptosis in GC-1spg cells via TUNEL assay. Results show that TUNEL-positive nuclei increased dose-dependently in ZEN treated cells. Scale bar = 100 μm. (**D**) The percentage of TUNEL-positive cells in each sample, represented as the mean ± SD of three independent experiments. (*n* = 3, ** *p* < 0.01 compared to the control).

**Figure 2 toxins-14-00148-f002:**
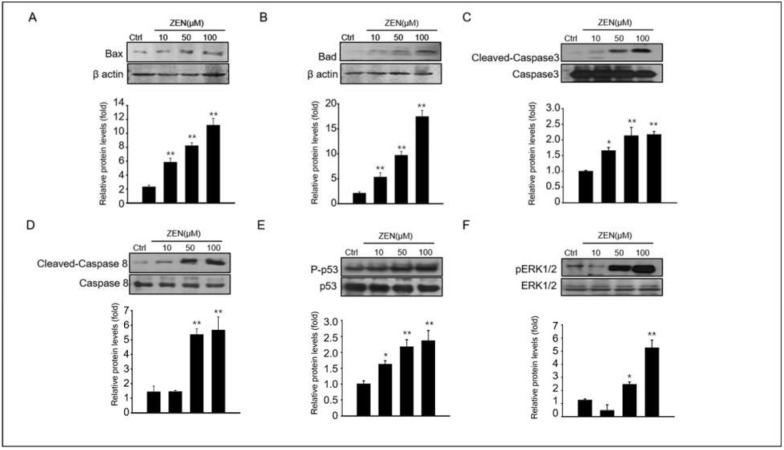
**Expression levels of pro-apoptotic protein on ZEN exposed GC-1 spg cells.** (**A**–**F**) Following treatment of 0–100 μM ZEN to GC-1 spg cells for 24 h, the Western blot technique was used to analyze the expression of pro-apoptotic proteins and signal molecules such as BAX, BAD, cleaved caspase-3, cleaved caspase-8, phosphor-p53, and phosphor-ERK1/2. The graph represents the densitometry analysis of each band normalized to that of each inactive protein band or β-actin. Values represent the mean ± SD of three independent experiments. (*n* = 3, * *p* < 0.05 and ** *p* < 0.01 compared to the control).

**Figure 3 toxins-14-00148-f003:**
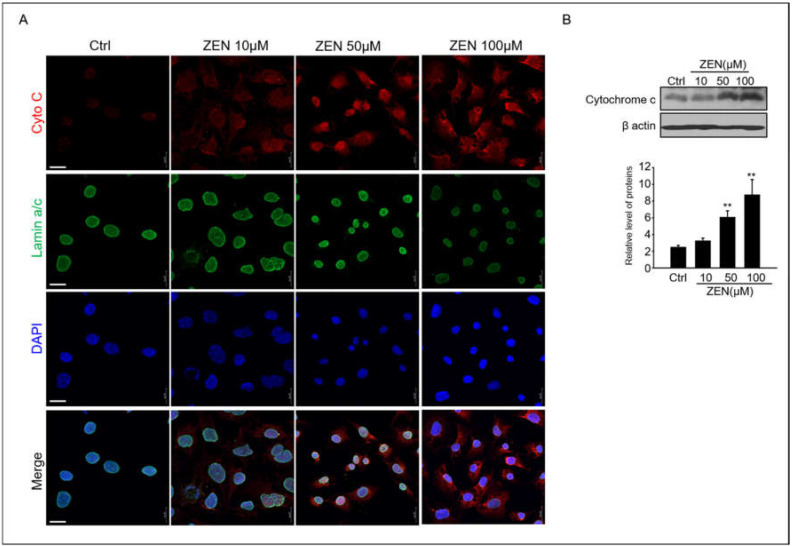
**ZEN induces cytochrome c release in GC-1 spg cells.** (**A**) GC-1 spg cells were treated with 0–100 μM ZEN for 24 h, and then immunostaining was performed with the cytochrome c antibody and co-labeled with Lamin A/C to visualize the nuclear envelope. DAPI was then utilized to visualize cell nuclei with confocal microscopy. Scale bar = 30 μm. (**B**) Western blot of cytochrome c in ZEN exposed GC-1 spg cells for 24 h. The density of cytochrome c bands was normalized to that of β-actin. Data are shown as the mean ± SD of three independent experiments. (*n* = 3, ** *p* < 0.01 compared to the control).

**Figure 4 toxins-14-00148-f004:**
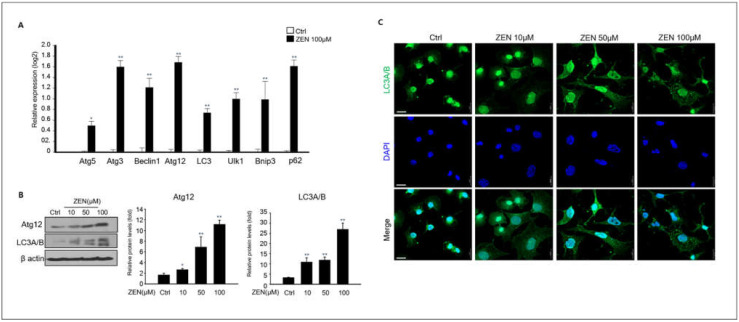
**Autophagy of GC-1 spg cell induced by ZEN**. (**A**) qPCR analysis of autophagy gene expressions Atg5, Atg3, Beclin 1, Atg12, LC3, Ulk1, Bnip3, and p62 in 0–100 μM ZEN exposed GC-1 spg cell for 24 h. Data represent mean ± SD. * *p* < 0.05, ** *p* < 0.01 compared to the control. (**B**) Western blot analysis of Atg2 and LC3A/B levels of ZEN exposed GC-1 spg cells. The graph shows the densitometric analysis of Atg2 and LC3A/B levels normalized to β-actin. Values represent the mean ± SD of three independent experiments. (*n* = 3, * *p* < 0.05 and ** *p* < 0.01 compared to the control). (**C**) Confocal immunofluorescence for LC3A/B in GC-1 spg cells following treatment of 0–100 μM ZEN and cell nuclei staining with DAPI.

**Table 1 toxins-14-00148-t001:** Primers used for reverse transcription–polymerase chain reactions.

Gene	Forward Primer	Reverse Primer
*Atg5*	5′-ACTTGCTTTACTCTCTATCAG-3′	5′-CATCTTCTTGTCTCATAACCT-3′
*Atg3*	5′-TCACAACACAGGTATTACAG-3′	5′-TCACAACACAGGTATTACAG-3
*Beclin1*	5′-GCGGGAGTATAGTGAGTT-3	5′-GGTGGCATTGAAGACATT-3
*Atg12*	5′-TAAACTGGTGGCCTCGGAAC-3′	5′-ATCCCCATGCCTGGGATTTG-3′
*LC3*	5′-CTTCGCCGACCGCTGTAA-3′	5′-GCCGGATGATCTTGACCAACT-3
*Ulk1*	5′-ACACACCTTCTCCCCAAGTG-3′	5′-GACGCACAACATGGAAGTCG-3′
*Bnip3*	5′-GCTCCTGGGTAGAACTGCAC-3′	5′-GCTGGGCATCCAACAGTATT-3′
*p62*	5′-GCACAGGCACAGAAGACAAG-3′	5′-CACCGACTCCAAGGCTATCT-3′
*Gapdh*	5′-GTCGGTGTGAACGGATTTG-3′	5′-CTTGCCGTGGGTAGAGTCAT-3′

**Table 2 toxins-14-00148-t002:** List of antibodies for immunostaining and immunoblotting.

1st Antibody	Company	Catalogue Number	Diluted
Atg12	Cell signaling	#4180	1:1500
LC3A/B	Cell signaling	#12741	1:1500
Caspase-8	Cell signaling	#9746	1:1500
Cleaved Caspase-8	Cell signaling	#8592	1:1500
BAX	Cell signaling	#14796	1:1500
BAD	Cell signaling	#9239	1:1500
P-p44/42 MAPK	Cell signaling	#4370	1:2000
P44/42 MAPK	Cell signaling	#9102	1:2000
p-p53	Cell signaling	#12571	1:1000
p53	Cell signaling	#2524	1:1000
Cytochrome c	Abcam	Ab133504	1:1000
β-Actin	Santa Cruz Biotech	SC-47778	1:1000

## Data Availability

The data presented in this study are available in this article.

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
