# Peer review of "Zearalenone Induces Apoptosis and Autophagy in a Spermatogonia Cell Line"

_toxins, 2022, doi:10.3390/toxins14020148_

Round 1

Reviewer 1 Report

Dear Authors,

The review manuscript entitled " Zearalenone induces apoptosis and autophagy in spermatogo- 2 nia cell line" for Toxins has been completed.

   The manuscript entitled " Zearalenone induces apoptosis and autophagy in spermatogo- 2 nia cell line " has demonstrates that ZEN has toxic effects on tGC-1 spg cells and induces both apoptosis and autophagy. In my opinion, the manuscript must be reject for journal. I declare that I have no competing interests. More detailed comments are as follows:

After reading the article, I searched many corresponding literatures on PubMed website and found that the author only changed the cell line to GC-1 SPG cell line. In view of the current situation of the previous research, the innovation of the article is worth considering, and it is recommended to reject the manuscript.

  1. Chen, F., Wen, X., Lin, P., Chen, H., Wang, A., & Jin, Y. (2019). HERP depletion inhibits zearalenone-induced apoptosis through autophagy activation in mouse ovarian granulosa cells. Toxicology letters301, 1–10. https://doi.org/10.1016/j.toxlet.2018.10.026
  2. Zhu, Y., Wang, H., Wang, J., Han, S., Zhang, Y., Ma, M., Zhu, Q., Zhang, K., & Yin, H. (2021). Zearalenone Induces Apoptosis and Cytoprotective Autophagy in Chicken Granulosa Cells by PI3K-AKT-mTOR and MAPK Signaling Pathways. Toxins13(3), 199. https://doi.org/10.3390/toxins13030199
  3. Wang, Y., Zheng, W., Bian, X., Yuan, Y., Gu, J., Liu, X., Liu, Z., & Bian, J. (2014). Zearalenone induces apoptosis and cytoprotective autophagy in primary Leydig cells. Toxicology letters226(2), 182–191. https://doi.org/10.1016/j.toxlet.2014.02.003

Author Response

[Reviewer #1]

Reviewer`s Comment :  The manuscript entitled " Zearalenone induces apoptosis and autophagy in spermatogo- 2 nia cell line " has demonstrates that ZEN has toxic effects on tGC-1 spg cells and induces both apoptosis and autophagy. In my opinion, the manuscript must be reject for journal. I declare that I have no competing interests. More detailed comments are as follows:

After reading the article, I searched many corresponding literatures on PubMed website and found that the author only changed the cell line to GC-1 SPG cell line. In view of the current situation of the previous research, the innovation of the article is worth considering, and it is recommended to reject the manuscript.

  1. Chen, F., Wen, X., Lin, P., Chen, H., Wang, A., & Jin, Y. (2019). HERP depletion inhibits zearalenone-induced apoptosis through autophagy activation in mouse ovarian granulosa cells. Toxicology letters301, 1–10. https://doi.org/10.1016/j.toxlet.2018.10.026
  2. Zhu, Y., Wang, H., Wang, J., Han, S., Zhang, Y., Ma, M., Zhu, Q., Zhang, K., & Yin, H. (2021). Zearalenone Induces Apoptosis and Cytoprotective Autophagy in Chicken Granulosa Cells by PI3K-AKT-mTOR and MAPK Signaling Pathways. Toxins13(3), 199. https://doi.org/10.3390/toxins13030199
  3. Wang, Y., Zheng, W., Bian, X., Yuan, Y., Gu, J., Liu, X., Liu, Z., & Bian, J. (2014). Zearalenone induces apoptosis and cytoprotective autophagy in primary Leydig cells. Toxicology letters226(2), 182–191. https://doi.org/10.1016/j.toxlet.2014.02.003

  • Our response: We thank for your comment.

In our opinion, Cytotoxic mechanism of same toxins can be differed with each cell types, and when an animal is exposed to a toxin, each organ in the body can react differently. GC-1 spg cell are type B spermatogonia, which undergo growth and become primary spermatocytes that undergo meiosis. In other word, GC-1 spg cell are early type of male germ cells. The study of mechanism of cytotoxicity of ZEN on early germ cell is critical for maintaining of reproductive function in male animals and human. We thought that these results of this study are meaningful that aspects which mentioned above.

Reviewer 2 Report

The article presents a properly designed experiment evaluating the toxic activity of ZEN in a cell line of spermatogonia. The implication of some apoptotic pathways was clearly demonstrated. The metodology was clear. I would suggest to include a brief section "Conclsuions".

  • Some Minor revisions

Title would better: “Zearalenone induces apoptosis and autophagy in a spermatogonia cell line”

Line 26 27: Articulate the sentence in this way: “Zearalenone (ZEN), also known as the F-2 toxin is a non-steroidal mycotoxin with estrogenic effects produced by some Fusarium and Gibberella species”

Line 27 – 29: Complete the sentence: “ZEN is mainly found in contaminated grains such as oats, corn, wheat, sorghum, and rice, but also in forages as well as in food and animal feed that are improperly stored [2, 3]” and add a reference.

Line 39-42: Modified the sentence: “Male mice exposed to this mycoestrogen showed an increased number of abnormal spermatozoa”

Line 72-74: I recommend deleting this “although other studies have reported the toxic effects of ZEN in the male reproductive system through in vivo studies, as mentioned above.”

Line 116 – 119: Alling this with the title of the figure in the line 120.

Line 172: Use Abbreviation ZEN

  • Include in the discussion a mention the issue that in vivo mixtures of ZEN and other dietary estrogens [such as ZEN-related metabolites, Alternaria-derived mycoestrogens (AHO, AME, TeA) and phytoestrogens] are co-occurring, affecting the same and to the here presented related pathways. Some literature on this issue:

Vejdovszky, K., Hahn, K., Braun, D., Warth, B., & Marko, D. (2017). Synergistic estrogenic effects of Fusarium and Alternaria mycotoxins in vitro. Archives of Toxicology, 91(3), 1447-1460.

Beszterda, M., & Frański, R. (2018). Endocrine disruptor compounds in environment: As a danger for children health. Pediatric Endocrinology, Diabetes & Metabolism, 24(2).

Ward, W. E., & Thompson, L. U. (2001). Dietary estrogens of plant and fungal origin: occurrence and exposure. In Endocrine Disruptors–Part I (pp. 101-128). Springer, Berlin, Heidelberg.

  • Write a short session of conclusions to clarify the main remarks of the study.

Author Response

[Reviewer #2]

Reviewer`s Comment: The article presents a properly designed experiment evaluating the toxic activity of ZEN in a cell line of spermatogonia. The implication of some apoptotic pathways was clearly demonstrated. The metodology was clear. I would suggest to include a brief section "Conclsuions".

â–¶Some Minor revisions (revised sentence expressed as “ red” in manuscripts)

Comment 1) Title would better: “Zearalenone induces apoptosis and autophagy in a spermatogonia cell line”

  • Our response: We thank and agree to Reviewer’s comment. We changed title to “Zearalenone induces apoptosis and autophagy in a spermatogonia cell line”

Line: 2

Comment 2) Line 26 27: Articulate the sentence in this way: “Zearalenone (ZEN), also known as the F-2 toxin is a non-steroidal mycotoxin with estrogenic effects produced by some Fusarium and Gibberella species”

  • Our response: We thank and agree to Reviewer’s comment. We edited sentence according to reviewer` comment.

Line: 36-37

Comment 3) Line 27 – 29: Complete the sentence: “ZEN is mainly found in contaminated grains such as oats, corn, wheat, sorghum, and rice, but also in forages as well as in food and animal feed that are improperly stored [2, 3]” and add a reference.

  • Our response: We thank and agree to Reviewer’s comment. We edited sentence according to reviewer` comment.

Line: 38-39

Comment 4) Line 39-42: Modified the sentence: “Male mice exposed to this mycoestrogen showed an increased number of abnormal spermatozoa”

  • Our response: We thank and agree to Reviewer’s comment. We edited sentence according to reviewer` comment.

Line: 51-52

Comment 5) Line 72-74: I recommend deleting this “although other studies have reported the toxic effects of ZEN in the male reproductive system through in vivo studies, as mentioned above.”

  • Our response: We thank and agree to Reviewer’s comment. We deleted that sentence

Line: 83

Comment 6) Line 116 – 119: Alling this with the title of the figure in the line 120.

  • Our response: Thanks for comment, we edited

Line: 107

Comment 7) Line 172: Use Abbreviation ZEN

  • Our response: Thanks for comment, we edited abbreviation to ZEN from ZEA in all Figure

Comment 7) Include in the discussion a mention the issue that in vivo mixtures of ZEN and other dietary estrogens [such as ZEN-related metabolites, Alternaria-derived mycoestrogens (AHO, AME, TeA) and phytoestrogens] are co-occurring, affecting the same and to the here presented related pathways. Some literature on this issue:

Vejdovszky, K., Hahn, K., Braun, D., Warth, B., & Marko, D. (2017). Synergistic estrogenic effects of Fusarium and Alternaria mycotoxins in vitro. Archives of Toxicology, 91(3), 1447-1460.

Beszterda, M., & FraÅ„ski, R. (2018). Endocrine disruptor compounds in environment: As a danger for children health. Pediatric Endocrinology, Diabetes & Metabolism24(2).

Ward, W. E., & Thompson, L. U. (2001). Dietary estrogens of plant and fungal origin: occurrence and exposure. In Endocrine Disruptors–Part I (pp. 101-128). Springer, Berlin, Heidelberg

  • Our response: Thank you for your comment, we added to discussion session according to your comment.

Line: 320-322

Comment 8) Write a short session of conclusions to clarify the main remarks of the study.

  • Our response: Thank you for your comment, we added short session of conclusions

Line: 337-340

Reviewer 3 Report

The article characterizes the mechanism of toxin action of ZEN on spermatogonia cell line in vitro. The methods are adequate. The manuscript is well written and the data are clearly presented.

However, the concentrations of ZEN used in the study should be discussed in the relation to the ZEN levels found in tissues and blood during natural and experimental intoxications.

I don’t agree with the sentence: “The molecular mechanisms underlying the toxic effects of the major mycotoxins have been established, and oxidative stress and the generation of free radicals have been implicated in mycotoxin toxicity [23, 24]”.

Please reverse microscopy and immunoblotting: “The expression and localization of cytochrome c 130 protein in GC-1 spg cells were observed using confocal immunofluorescence microscopy 131 and immunoblotting, respectively.”

Author Response

[Reviewer #3]

Reviewer`s Comment: The article characterizes the mechanism of toxin action of ZEN on spermatogonia cell line in vitro. The methods are adequate. The manuscript is well written and the data are clearly presented. (revised sentence expressed as “ blue” in manuscripts)

However, the concentrations of ZEN used in the study should be discussed in the relation to the ZEN levels found in tissues and blood during natural and experimental intoxications

  • Our response: Thank you for your comment, we added information which you mentioned

Line: 315-319

I don’t agree with the sentence: “The molecular mechanisms underlying the toxic effects of the major mycotoxins have been established, and oxidative stress and the generation of free radicals have been implicated in mycotoxin toxicity [23, 24]”.

  • Our response: Thank you for your comment, we edited that sentence

Line: 72-74

Please reverse microscopy and immunoblotting: “The expression and localization of cytochrome c 130 protein in GC-1 spg cells were observed using confocal immunofluorescence microscopy 131 and immunoblotting, respectively.”

  • Our response: Thank you for your comment, we edited that sentence according to your comment.

Line: 169-170

Round 2

Reviewer 1 Report

After a round of revision, I carefully read the author's revised paper and published literature. I think this article can be accepted for publication.